# Optimization of Sb_2_S_3_ Nanocrystal Concentrations in P_3_HT: PCBM Layers to Improve the Performance of Polymer Solar Cells

**DOI:** 10.3390/polym13132152

**Published:** 2021-06-29

**Authors:** E. M. Mkawi, Y. Al-Hadeethi, R. S. Bazuhair, A. S. Yousef, E. Shalaan, B. Arkook, A. M. Abdeldaiem, Rahma Almalki, E. Bekyarova

**Affiliations:** 1K.A.CARE Energy Research and Innovation Center, King Abdulaziz University, Jeddah 21589, Saudi Arabia; 2Department of Physics, Faculty of Science, King Abdulaziz University, Jeddah 21589, Saudi Arabia; yalhadeethi@kau.edu.sa (Y.A.-H.); eshalan@kau.edu.sa (E.S.); barkook@kau.edu.sa (B.A.); abdeldaiem@hotmail.com (A.M.A.); ralmalki0259@stu.kau.edu.sa (R.A.); 3Department of Drawing and Art, College of Art and Design, Jeddah University, Jeddah 23345, Saudi Arabia; Rehabsyb@gmail.com (R.S.B.); Asyuosef@uj.edu.sa (A.S.Y.); 4Department of Chemistry, University of California at Riverside, Riverside, CA 92521, USA; elena.bekyarova@ucr.edu

**Keywords:** solar cells, Sb_2_S_3_ nanocrystals, P_3_HT: PCBM polymer

## Abstract

In this study, polymer solar cells were synthesized by adding Sb_2_S_3_ nanocrystals (NCs) to thin blended films with polymer poly(3-hexylthiophene)(P_3_HT) and [6,6]-phenyl-C61-butyric-acid-methyl-ester (PCBM) as the p-type material prepared via the spin-coating method. The purpose of this study is to investigate the dependence of polymer solar cells’ performance on the concentration of Sb_2_S_3_ nanocrystals. The effect of the Sb_2_S_3_ nanocrystal concentrations (0.01, 0.02, 0.03, and 0.04 mg/mL) in the polymer’s active layer was determined using different characterization techniques. X-ray diffraction (XRD) displayed doped ratio dependences of P_3_HT crystallite orientations of P_3_HT crystallites inside a block polymer film. Introducing Sb_2_S_3_ NCs increased the light harvesting and regulated the energy levels, improving the electronic parameters. Considerable photoluminescence quenching was observed due to additional excited electron pathways through the Sb_2_S_3_ NCs. A UV–visible absorption spectra measurement showed the relationship between the optoelectronic properties and improved surface morphology, and this enhancement was detected by a red shift in the absorption spectrum. The absorber layer’s doping concentration played a definitive role in improving the device’s performance. Using a 0.04 mg/mL doping concentration, a solar cell device with a glass /ITO/PEDOT:PSS/P_3_HT-PCBM: Sb_2_S_3_:NC/MoO_3_/Ag structure achieved a maximum power conversion efficiency of 2.72%. These Sb_2_S_3_ NCs obtained by solvothermal fabrication blended with a P_3_HT: PCBM polymer, would pave the way for a more effective design of organic photovoltaic devices.

## 1. Introduction

Solar cell-based organic semiconductors have many advantages, such as low cost, lightweight, flexibility, low material consumption, easy fabrication, and large area production [1,2].The organic photovoltaic (PV) solar cells continue to find a widespread application, in particularly in the following areas: solar farms, remote locations, powering stand-alone devices, powering earth-orbiting satellites and space stations, building structures such as windows and roof tiles, martial utilizations, and in aviation to power aircrafts at high altitudes. Polymers are often used as adjustment additives in dye-sensitized solar cells to provide a desirably resilient substrate, a frame structure of the semi solid state electrolytes, in addition to the pore/film formation in photoanode films. Moreover, polymers are added to enhance a solar cell device’s performance, such as reinforcing the processes of crystallization and nucleation in the perovskite solar cell films. Polymers are applied as buffer layers or donor layers to improve a device’s efficiency. Polymers are also applied as electron transmitters, hole transmission materials, as well as interfacial layers, which improve the carrier separation efficacy and minimize the recombination.

Poly(3-hexylthiophene) and [6,6]-phenyl-C61-butyric acid methyl ester, P_3_HT-PCBM blends are promising organic polymers that have been used as photovoltaic materials [3]. They are considered to be a promising fullerene derivative-based donor-acceptor electron material for organic solar cells. PCBM polymers are fullerene derivatives as electron acceptors for organic photovoltaics because of their high electron mobility. P_3_HT is a member of the polythiophene-conducting polymer family, in which excitation of the π-orbit electron in P_3_HT produces photovoltaic effects in the blend [4]. The blend’s energy gap is approximately 1.8 eV and should exhibit a high absorption wavelength around 650 nm.

The lifetimes of organic solar cells remain short due to the degradation mechanisms that occur in organic compounds and oxidation of the electrode materials. However, strictly controlling this morphology is severely limited, which will lead to charge carrier recombinations because of incomplete pathways for both types of charge carriers if the films are thicker than 150 nm [5]. The donor–acceptor morphology can be controlled using the metal sulfide geometry, which can create good percolation pathways. Using ordered nanostructures such as nanocrystals or nanoparticles is promising for controlling the final structure, as electrons can be transported along one-dimensional structures over many micrometers, which reduces carrier recombinations [5,6]. P_3_HT forms ordered microcrystalline structures in the solid state. The presence of ordered crystalline structures in solid thin films helps to obtain high device performance because of the improved hole mobility from stacking thiophene rings and forms an enhanced light absorption with ordered structures in longer wavelength regions. Thus, incorporating inorganic and nanostructure semiconductors has several advantages, such as high electron mobility and physical and chemical stability [7,8]. However, some disadvantages are present, such as the formation of large aggregates of nanostructures, which may be effective in active layer morphology charge mobility.

Many nanostructure materials, such as PbS, ZnO, CdS, TiO_2_, and Sb_2_S_3_, have been used in polymer solar cells due to their high electron mobility. Sb_2_S_3_ is a semiconductor material with a unique one-dimensional crystal structure, higher stability in air, and diverse Sb-S bond lengths. Sb_2_S_3_ is a non-toxic, abundant material with an indirect band gap of 1.7–1.8 eV, an absorption coefficient higher than 104 cm^−1^, and high electric conductivity, which make it a suitable material for use as light harvesters in photovoltaic applications. Sb_2_S_3_ nanocrystals may provide a significant contribution to the absorption in polymer/Sb_2_S_3_ nanocrystal solar cells [9,10,11]. In a study incorporating semiconductor P_3_HT:PCBM nanoparticles, Zhao et al. [12] showed that adding PbS quantum dots (QDs) to P_3_HT changed the chemical structure, which improved the active layer via an optimized phase separation and increased carrier transfers. Kim et al. [13] reported a PCE of approximately 2.98% by combining various concentrations of ZnO nanoparticles grown using the hydrothermal method with a P_3_HT: PCBM blend. They suggested that the improved charge balance and performance may have been due to reducing the charge recombinations and oxygen vacancies from the cathodes.

In this study, we report the fabrication of Sb_2_S_3_ NCs for their application in one of the most efficient organic solar cells consisting of PTB7: PCBM organic photovoltaic blends. To the best of our knowledge, the effect of the Sb_2_S_3_ NC additive on the properties of P_3_HT:PCBM blend solar cells that employ solid hole conductors has not been investigated. High-quality Sb_2_S_3_NPs were synthesized using solvothermal techniques at a hydrothermal temperature of 180 °C. The effect of different Sb_2_S_3_ NC concentrations incorporated with PTB7: PCBM was investigated in detail using different characterization methods. The experimental results indicated that the self-assembly of the PTB7: PCBM Sb_2_S_3_ NC polymer and intermolecular orientation in the P_3_HT crystallite was greatly influenced by Sb_2_S_3_ nanocrystal doping. By adding Sb_2_S_3_ NCs, the electrons and holes transfer in the polymer active layer enhanced our results, which led to an improved carrier separation efficiency and a reduced recombination. The absorption spectra of the sample, after adding Sb_2_S_3_ NCs, was better attributed to the π-π* transitions. The Sb_2_S_3_NPs help to disperse the P3HT chains in a solution and promote transformation and, thus, improve the crystallization of P3HT (during film forming process) as well as the device’s efficiency. The solar cell device in a structure glass/ITO/PEDOT:PSS/P_3_HT: PCBM:Sb_2_S_3_:NCs/MoO_3_/Ag achieved a maximum power conversion efficiency of 2.72% using an Sb_2_S_3_NC concentration of 0.04 mg/mL. Hence, Sb_2_S_3_NC-doped P3HT-PCBM thin films have great potential applications as active layers in solar cell devices and can be an efficient method to improve thin film properties.

## 2. Experiment

### 2.1. Sb_2_S_3_ Nanocrystals Fabrication

For the typical solvothermal fabrication of Sb_2_S_3_ nanocrystals, 0.2 mmol of SbCl_3_ and 4 mmol of I-cystine were dissolved in 10mL of oleylamine. Then, 4 mmol of thiourea was dissolved in 10 mL of oleylamine in a separate beaker under stirring for 2 h. The antimony solution was dropped slowly into the latter solution under vigorous magnetic stirring. The mixed solutions were then transferred into a 50-milliliter Teflon-lined, stainless steel autoclave and placed in a tubular furnace at 180 °C for 24 h. The resulting precipitate was first centrifuged, and the black product was washed with ethyl alcohol and deionized water and dried at 70 °C for 5 h.

### 2.2. Device Fabrication

Indium tin oxide (ITO)-coated glass substrates, approximately 120 nm thick with a sheet resistance of around 15 Ω/sq and a 1.5 × 1.5 cm^2^ device area were used as an anode contact in the organic solar cell device. Acetone or isopropanol were soaked in an ultrasonic bath for 5 min, followed by drying in an N_2_-filled glove box. A PEDOT:PSS polymer was deposited at 3500 rpm for 50 s, using spin coating followed by annealing at 120 °C for 15 min in a furnace in air, resulting in a thickness of ∼60 nm. The PEDOT:PSS/ITO/glass substrates were moved to a nitrogen-supplied glove box and annealed again at 130 °C for 10 min to avoid humidity. The blended P_3_HT and PCBM solution was prepared by dissolving the polymers in 1,2-dichlorobenzene with a ratio of 1:1 and 50 mg of each polymer. The mixed solutions were stirred at 60 °C overnight. The Sb_2_S_3_ nanocrystals were blended in the P_3_HT: PCBM in a concentration range of 0.01, 0.02, 0.03, and 0.04 mg/mL. The P_3_HT:PCBM:Sb_2_S_3_:NC nanocomposites were spin-coated on top of PEDOT:PSS layers at 1500 rpm for 40 s inside a glove box. The active layer’s average thickness was 100–120 nm, and the spin-coated layer was annealed at 140 °C for 10 min in a nitrogen-filled glove box. A buffer layer, with approximately 20 nm of MoO_3_, was deposited using RF spurting. The thick Ag electrode (70 nm) was thermally evaporated, and the active area was approximately 0.1 cm^2^. The final device had the following structure: ITO/PEDOT:PSS/P_3_HT:PCBM:Sb_2_S_3_:NCs/MoO_3_/Ag. A schematic illustration of a typical device structure is shown in Figure 1.

### 2.3. Characterization

The structural properties of the Sb_2_S_3_ nanocrystals and P_3_HT:PCBM:Sb_2_S_3_ NCs were investigated using an X-ray diffractometer (MiniFlex, Rigaku, Tokyo, Japan) with monochromatic Cu Kα radiation (λ = 1.5405 Å). The UV–visible spectroscopy investigation was performed using a Varian Cary 100 spectrophotometer (Agilent Technologies, Santa Clara, California, USA). A transmission electron microscopy (TEM) analysis was obtained using a JEOL 2010F (JEOL 2010F field emission high resolution scanning/transmission electron microscope, Akishima, Tokyo, Japan) to examine the Sb_2_S_3_ nanocrystal’s properties. Atomic force microscopy (AFM) measurements were performed on a standard Keysight 5500 scanning probe microscope (Keysight Technologies Fountain grove Parkway, Santa Rosa, CA 95403) in the intermittent contact mode in air. The photoluminescence spectra were collected using a Hitachi F-7000 spectrometer (Hitachi F-7000 fluorescence spectrophotometer, Toranomo, Minato-ku, Tokyo, Japan) equipped with a red-sensitive detector. The Fourier-transform infrared spectroscopy (FTIR) of the polymer/NC blended films was investigated using a Nicolet 8700 (Thermo Fisher Scientific, Madison, WI, USA) spectrometer. Raman spectroscopy of the active layer was performed using a Renishaw inVia Raman microscope (Renishaw inVia Raman microscope, Gloucestershire, United Kingdom) (λ = 514 nm). The current density–voltage (J–V) characteristics of the polymer device were measured under AM 1.5 G and 100 mW cm^2^ illumination using a Keithley 2400 (Keithley, Tektronix, Solon, Ohio, USA) for source measurement.

## 3. Results and Discussion

The crystal sizes and phase information on the Sb_2_S_3_ were confirmed using X-ray diffraction (XRD) patterns from the pure Sb_2_S_3_ nanocrystals prepared using the solvothermal method as shown in Figure 2a. All of the XRD patterns in Figure 2a present the stibnite structure of the Sb_2_S_3_ (JCPDS No. 42-1393). For example, the diffraction peaks at 2θ = 17.7, 24.4, 32.8, 35.1, 44.1, 54.3, and 46.4° corresponded to the (120), (130), (221), (301), and (511) orientations, respectively, with a preferred orientation along the (130) plane. The sharp and high-intensity peaks indicated that the product had high crystallinity. No peaks were associated with other phases, which indicated the sample’s high purity. The lattice parameter identical to the orthorhombic type was a = 11.23 Å, b = 11.28 Å, and c = 3.83 Å. These results agreed with previously reported studies of Sb_2_S_3_ materials [14,15]. The crystallite size of the Sb_2_S_3_ nanocrystals was 41 nm, as determined using the Scherrer formula [16].

To assess the crystalline development of the P_3_HT:PCBM: Sb_2_S_3_ NCs’ active layer, the sample’s XRD patterns were recorded. Figure 2b shows the XRD profiles of P_3_HT:PCBM: Sb_2_S_3_ NC blended films spin-coated at 1500 rpm for 40 s. The XRD analysis was recorded in a narrow range (2θ = 3–10°). The increase in the (100) peak intensity corresponding to P_3_HT was observed at 2θ = 5.4°, which agreed with prior studies [17,18]. The diffraction peak located at 2θ = 19° corresponded to crystalline PCBM [19]. The XRD (100) peaks shifted to lower angles, from 5.58 to 5.39°, as the dispersion degree increased. This shift (change in D spacing) indicated the improved diffusion of the PCBM into the P_3_HT, decreasing the distance between them [20]. The diffraction peaks became narrower and sharper as the Sb_2_S_3_NCs’ concentration increased, indicating an increase in the crystallinity, suggesting orderliness, and increasing the intermolecular π plane. The P_3_HT:PCBM: Sb_2_S_3_ NCs’ blend displayed a high crystallinity that improved regardless of the presence of fullerene. The growth of the P_3_HT aggregate improved the chains’ crystallinity and hole mobility, which resulted in positive J–V curves [21].

Raman spectroscopy in a range of 250 to 2500 cm^−1^ was used to investigate the molecules’ irrational mode in the P_3_HT:PCBM:Sb_2_S_3_ NC blended films. The Raman spectra of the P_3_HT:PCBM: Sb_2_S_3_ NC blended films shown in Figure 3 features the vibration modes, as reported in previous studies, including the vibration spectra, and the stretching and bending modes with different relative intensities varying from high to low values. The amount of Sb_2_S_3_ NCs contributed to the changes in the peak intensity. The peaks at ∼578 to ∼782 cm^−1^ coincided with C-H out-of-phase bending [22]. As most studies of polymers suggested, the peak located at 728 cm^−1^ was related to rocking vibrations in the C-S-C thiophene ring of the P_3_HT [22,23]. The peak found at ∼1086 cm^−1^ was not from the P_3_HT molecules; therefore, this vibration peak may have corresponded to interactions of the P_3_HT molecules and the Sb_2_S_3_ NCs. The peak located at ∼1378 cm^−1^ was associated with the asymmetric vibrations of C=C skeletal stretching deformation, and the peak appearing at∼1518 cm^−1^ corresponded to stretching vibrations from the P_3_HT, which agrees with a previous study [24]. A higher intensity peak at ∼1442 cm^−1^ was related to the high P_3_HT structural order [25]. At this peak, the combined skeletal stretching of the complete chain, or at least a large part of it, was found. This mode was dominated by the inter-ring C-C stretching vibration mode and was the origin of the vibronic structure of the absorption and emission spectra [25,26]. The intensity of the ∼1442 cm^−1^ peak related to the phonon features was reduced due to the presence of the Sb_2_S_3_ NCs, suggesting an increased order in the blended system and enhanced conjugation lengths.

To obtain accurate morphological and size information, TEM images and diffraction patterns were obtained for the sample. Figure 4a–c show TEM images of the Sb_2_S_3_ nanocrystals prepared using the solvothermal method at 230 °C. The typical TEM images shown in Figure 4a confirm that the morphology of the Sb_2_S_3_ was a nanocrystalline along the (001) direction with a diameter of approximately 100 nm. The selected area electron diffraction SAED images in Figure 4b show the sharp spots of the Sb_2_S_3_, where the diffraction spots indicate the fully crystalline nature of the Sb_2_S_3_NCs and the main diffraction plane matched with the standard XRD patterns shown in Figure 2. The high-resolution TEM (HRTEM) images are shown in Figure 4c and the lattice spacing of 0.32 nm agrees well with the (130) diffraction of the Sb_2_S_3_. The SAED images and corresponding HRTEM images demonstrated that the crystalline nature of the Sb_2_S_3_ NCs agreed with the XRD and Raman results.

The optical absorption spectra of the Sb_2_S_3_ nanocrystals recorded in a range of 200—800 nm is shown in Figure 5a. The Sb_2_S_3_ nanocrystals had high absorption coefficients above 5 × 10^4^ cm^−1^ in a wavelength range of 350–750 nm. The strong absorption intensity was attributed to the good quality of the Sb_2_S_3_ nanocrystals. A large absorption coefficient is important for solar cell applications, which implies a high short-circuit current density. The energy gaps of the Sb_2_S_3_ NCs were calculated by plotting the αhv^1/2^ vs. hv and extrapolating the linear portion of the curve to αhv = 0, as shown in Figure 5a (insert). The optical bandgap (Eg) determined using the Tauc equation [27] was 1.79 eV.

To investigate the photon-gathering ability of the P_3_HT:PCBM:Sb_2_S_3_ NCs’ active layer, UV–visible absorption spectra of the sample were recorded in a wavelength range of 300–800 nm. Figure 5b shows the UV–visible absorption spectra of the P_3_HT:PCBM:Sb_2_S_3_ NC blended films prepared using different concentrations of Sb_2_S_3_ NCs (0.01, 0.02, 0.03, and 0.04 mg/mL). The absorption spectra in a range of 450–600 nm were attributed to the main P_3_HT polymer. The absorption spectra of the P_3_HT:PCBM:Sb_2_S_3_NCs’ active layer showed an absorption peak of π-π* aggregate formation at a wavelength of 512 nm, with two small shoulders at 550 and 604 nm. The P_3_HT polymer displayed an atypical absorption band at approximately 512 nm that was attributed to the π-π* transitions, this result is in good agreement with previous studies [28,29]. The π-π* band of the active layer showed a 3-nanometer red shift corresponding to the P_3_HT polymer, suggesting more efficient π stacking. Sb_2_S_3_ NCs had absorption spectra below 550 nm in previous studies. Sb_2_S_3_ NC doping may have strongly contributed to the buffer absorbance within a 300–550 nm range. Doping resulted in a clear red shift of the optical absorption, mainly in the 512 nm band, which shifted from 500 to 512 nm. The red shift in the absorption band could be attributed to increasing the π electron delocalization, lowering the energy band of the π and π*, and improving the optical π-π* transitions [30,31]. The increase in the light-harvesting properties of the P_3_HT:PCBM:Sb_2_S_3_ NC blend improved the photo-generated carriers and enhanced the charge transport due to the π-π interactions between the Sb_2_S_3_ NCs and the P_3_HT molecules. The absorption strength was increased, along with the different concentrations, due to the improved polymer crystallinity. However, the improved absorption may have been due to the decreased film roughness, which affected the light scattering in the blended films, resulting in increased absorption in the active layer.

Figure 6 shows a schematic energy level diagram of the energy and charge transfer effects of the P_3_HT:PCBM:Sb_2_S_3_ NC blend. There are the following three possible reasons for the electron movement in the P_3_HT:PCBM:Sb_2_S_3_ NC system: electron transport from the P_3_HT to the PCBM; electron transfer from the P_3_HT to the Sb_2_S_3_ NCs; and electrons moving from the P_3_HT to the Sb_2_S_3_ NCs, and then to the PCBM. In these cases, the electrons transformed into Ag, while the holes transformed out of the P_3_HT to the ITO layer, which reduced the chances of carrier recombination. Thus, incorporating the Sb_2_S_3_ NCs with the polymer contributed to more photo-induced charge carrier separations/transfers that increased the photo-generated exciton dissociation [8,32]. The processes can occur depending on the excitation energy. When excitons are formed upon light absorption in the Sb_2_S_3_ NCs, it is expected that the electrons will be transferred to PCBM and holes to P3HT. When excitations are generated in P3HT, we can predict from the energy levels that an electron transfer will occur toward the Sb_2_S_3_ NCs and/or PCBM [33]. To learn more about the processes occurring between the blend components, we investigated steady state and time-resolved PL.

To study the effect of the Sb_2_S_3_ NCs on the exciton dissociation, the PL spectra of the P_3_HT:PCBM:Sb_2_S_3_ NC thin films with different concentrations of Sb_2_S_3_ NCs were investigated using a PL spectral system, as shown in Figure 7. The PL peak of the P_3_HT:PCBM:Sb_2_S_3_ NCs was observed at 630 nm, in good agreement with previous studies for P_3_HT:PCBM [34,35]. At higher Sb_2_S_3_ NC concentrations, the peaks become more intense and emission peaks at 635nm red-shifted to 639 nm, which may have been related to band-to-band emission in the P_3_HT, since it had a narrow band gap of approximately 1.9 eV.

The increased PL intensity indicated the improved phase separation size, which benefited the charge transport and collection. The size range of the P_3_HT and PCBM exceeded the exciton diffusion length because of the increased phase separation between the PCBM and P_3_HT after doping. A large increase in the PL intensity was observed, consistent with the increased diffusion of the PCBM in the P_3_HT matrix, leading to increased carrier transfers [36]. Consequently, the exciton-dissociation efficiency decreased, while the PL efficiency increased. These results indicated that the energy and charge transfer occurred between the P_3_HT, PCBM, and Sb_2_S_3_ NC material.

Atomic force microscopy (AFM) was used to investigate the tapping mode operation and phase distribution of the blends. Figure 8a–d shows AFM height images of the surface roughness and grain size of the P_3_HT:PCBM:Sb_2_S_3_ NC thin films prepared with different Sb_2_S_3_ NC concentrations. The images show the network structures of self-organized P_3_HT chains similar to the seed-like polymer chains reported in previous studies [37,38]. The bright areas (higher phases) in the phase images can be identified as P_3_HT-rich regions, while the dark areas (lower phases) can be identified as PCBM-rich regions. The phase images of the active layer display a considerable variation in the phase segregation length scale. The size of the P_3_HT area and PCBM domains, in a range of 24.3 to 59.27 nm, depended on the NC concentration. This indicated that two sequential polymer chains were separated within the exciton diffusion length, with PCBM nanoclusters established between the chains.

Evidently, the Sb_2_S_3_ NC acted as a compatibilizer and modified the average domain size of the PCBM by the intermolecular hydrogen bonds generated from the C-H-Os bonds; therefore, the active layer became smoother. The increase in the surface roughness could also have been due to the improved crystallinity of the P_3_HT in the film. This occurred because of the decreased P_3_HT aggregation size and increased number of P_3_HT single chains. The P_3_HT crystallization degree and effective phase blending had a major influence on the electron and hole mobility in the photovoltaic blend. The P_3_HT crystallization degree and effective phase blending had a major influence on the electron and hole mobility, which enhanced the carrier recombination and reduced the recombination layer of the solar cell device. As reported in prior studies, a high-efficiency photovoltage has a high surface roughness in P_3_HT:PCBM blended thin films [39].

Figure 9 displays the Fourier transform infrared (FT-IR) spectra recorded in the 250–4000 cm^−1^ spectral regions in the P_3_HT:PCBM:Sb_2_S_3_ NC blended films. In the 2700–3100 cm^−1^ spectral region, the methylene and methyl stretching vibration bands from the hexyl side chains of the thiophene rings were notable. The P_3_HT polymer exhibited vibration bands at 2954 and 3054 cm^−1^ corresponding to the stretching and asymmetric vibrations of =C-H and C=C [40,41]. The band at 1260 cm^−1^ corresponded to the dipole-derivative vector perpendicular to the ring plane, and the band near 1048 cm^−1^ was due to C-H in-plane bending [42]. The vibration mode at 1454 cm^−1^ was due to the deformation vibrations of the CH and CH_3_ [41]. The vibration modes in the 1600–1800 cm^−1^ region, such as 1715 cm^−1^, were due to photo-degradation products, including C=O groups [43]. The vibration mode at 1107 cm^−1^ corresponded to O=S stretching due to sulfonic esters [44]. The band near 820 cm^−1^ was related to the asymmetric deformation of CH3 vibrations (aromatic out-of-plain vibrations) [45]. Finally, the band at 1591cm^−1^ was due to C=C bands [46].

Our results were in good agreement with previous studies of FT-IR vibrations of pristine P_3_HT:PCBM. Very considerable changes suggested chemical degradation, as demonstrated after doping, leading to an essential increase in the peaks’ intensity in the vibration bands at 820, 2954, 2363, and 1260 cm^−1^, features of the hexyl side chains. This indicated that the hexyl side chains began to separate from the thiophene rings and eventually volatilized. As a result, as the doping concentration increased, the thiophene rings strengthened, affecting the increase in the P_3_HT band’s intensity, with a small shift related to the improved polymer chains leading to the phase separation of the P_3_HT:PCBM blend. The FT-IR results described the phase separation processes in the blend and vibration modes of the polymer chemical groups.

To evaluate the influence of the Sb_2_S_3_ NCs on the performance of theP_3_HT:PCBM:Sb_2_S_3_ NCs’photovoltage, four polymer devices with different concentrations of Sb_2_S_3_ NC-doped active layers (displayed in Figure 10) were fabricated. Figure 10 shows the current–voltage (IV) curves of the devices under AM 1.5G simulation and 100 mW/cm^2^ illumination. Table 1 summarizes the results of the open-circuit voltage (V_oc_), short-circuit current (I_sc_), fill factor (FF), and solar energy conversion efficiency (η) of the devices obtained with differentSb_2_S_3_ NC concentrations. The IV curves display the increases in the photocurrent that corresponded to the presence of theSb_2_S_3_ NCs in the active layer. In reference to the doping concentration of the Sb_2_S_3_ NCs shown in Table 1, 0.04 mg/mL was the optimum weight concentration that produced the highest efficiency η = 2.72% related to I_sc_ = 10.04 mA/cm^2^, V_oc_ = 412 mV, and FF = 66%.

Generally, the increase in the FF and V_oc_ was due to the increase in the R_sh_ and decrease in the R_s_. The device-resistant reduction was primarily attributed to the improved morphology in the active layer. The significantly increased J_sc_ and FF may have been due to a new network of Sb_2_S_3_ NCs that facilitated electron transport in the polymer’s active layer and a small range of PCBM aggregation that decreased the carrier recombination losses and increased the current density. In addition, the improved photocurrent corresponded to enhanced entrapment and light absorption, which were demonstrated by an energetic disorder or the improved crystallinity of the P_3_HT. The increase in the J_sc_ may also have been due to the increased incorporation of the Sb_2_S_3_ NCs in the P_3_HT, which demonstrated that the Sb_2_S_3_ NCs could work as electron acceptors comparable to the PCBM, which helped separate the bound photo-generated excitons. The decreases in V_oc_ with a lower concentration of Sb_2_S_3_ may be due to the weak interaction that occurs between the polymer and the nanocrystals’ interface, which leads to the incomplete charge transfer between the polymer and the nanocrystals, that could decrease the transport of charge-carriers in the hybrid solar cell. This mechanism could have caused many free electrical charges, increased the short-circuit current density, and thus improved the power conversion efficiency. Thus, Sb_2_S_3_ NCs blended in a polymer matrix provide a large interfacial area for fast charge dissociation at the interface to ensure that a maximum number of charge carriers contribute to improve the short current density. The performances of fabricated devices can be improved further by incorporating a thin layer of MoO_3_ as a hole transport layer between the hybrid blend and the back electrodes’ Ag.

## 4. Conclusions

In conclusion, we successfully synthesized highly crystalline Sb_2_S_3_ NCs using the solvothermal method. We synthesized bulk heterojunction organic solar cells based on P_3_HT:PCBM:Sb_2_S_3_ NCs by adding Sb_2_S_3_ NCs to the active layer. The device’s electrical, morphological, and optical properties were significantly affected by the Sb_2_S_3_ NC concentration in the P_3_HT:PCBM. The doping concentration improved the surface roughness of the active layer and tapping mode operation. The phase distribution of the blends was also investigated. The high crystalline polymer enhanced the red shift of the optical absorption, increased the photoluminescence intensity, and narrowed the full width at half the maximum of the Raman peaks. The results indicated that the Sb_2_S_3_ NCs strongly affected the flat-on orientation, which increased the charge carrier transport assisted by the π-π interactions. The increased number of P_3_HT single chains and the phase separation increased the free electrons, which effected the absorption and mobility and reduced the charge recombination in the active layer blend prepared using a concentration of 0.04 mg/mL. Our studies suggest that the charge separation and current generation in P_3_HT:PCBM:Sb_2_S_3_:NC-based devices result mainly from Sb_2_S_3_ NCs’ light absorption and subsequent hole-transfer from the inorganic semiconductor to the organic hole transporting material. The best power conversion efficiency (PCE) of the polymer solar cells was 2.72%, using a glass/ITO/PEDOT:PSS/P_3_HT:PCBM:Sb_2_S_3_:NC/MoO_3_/Ag device structure, under AM 1.5 sun, and global irradiation of 1000 W m^−2^ by blending the Sb_2_S_3_ NCs with a concentration of 0.04 mg/mL in the active layer. An expected mechanism is also proposed to explain the superior performance of Sb_2_S_3_NC-doped P_3_HT:PCBM at optimal content.

## Figures and Tables

**Figure 1 polymers-13-02152-f001:**
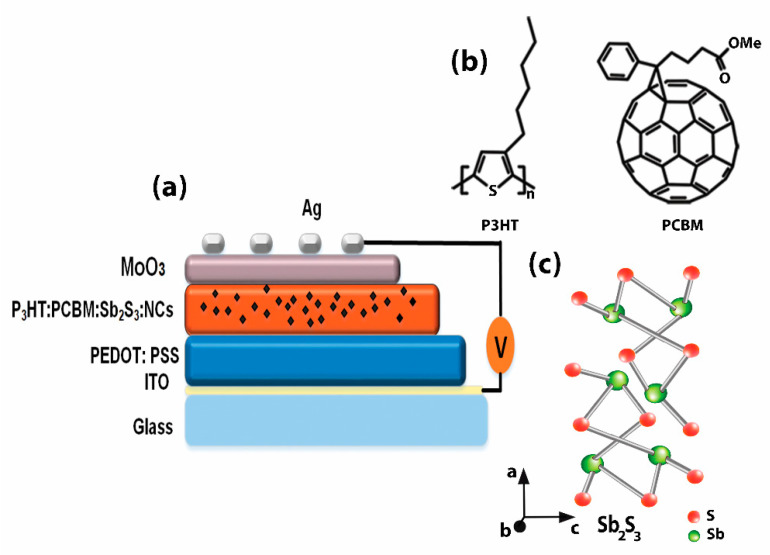
(**a**) Schematic configuration of the inverted polymer solar cell showing the glass/ITO/PEDOT:PSS/P_3_HT:PCBM:Sb_2_S_3_:NCs/MoO_3_/Ag layers and chemical structure of the (**b**) P_3_HT, PCBM, and (**c**) Sb_2_S_3_ nanocrystals.

**Figure 2 polymers-13-02152-f002:**
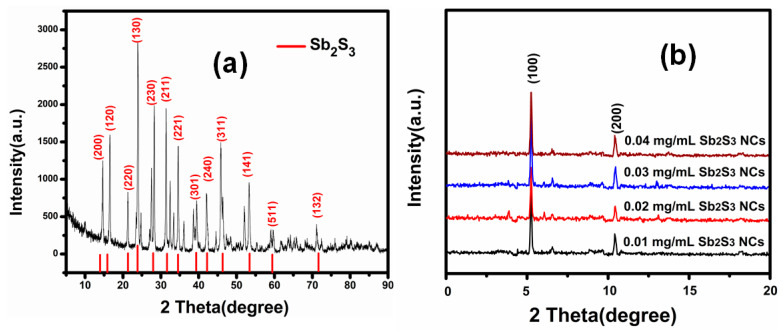
XRD diffraction of (**a**) pure Sb_2_S_3_ nanocrystals prepared using the solvothermal method (**b**) in P_3_HT:PCBM:Sb_2_S_3_ NC blended films spin-coated with different Sb_2_S_3_ NC concentrations.

**Figure 3 polymers-13-02152-f003:**
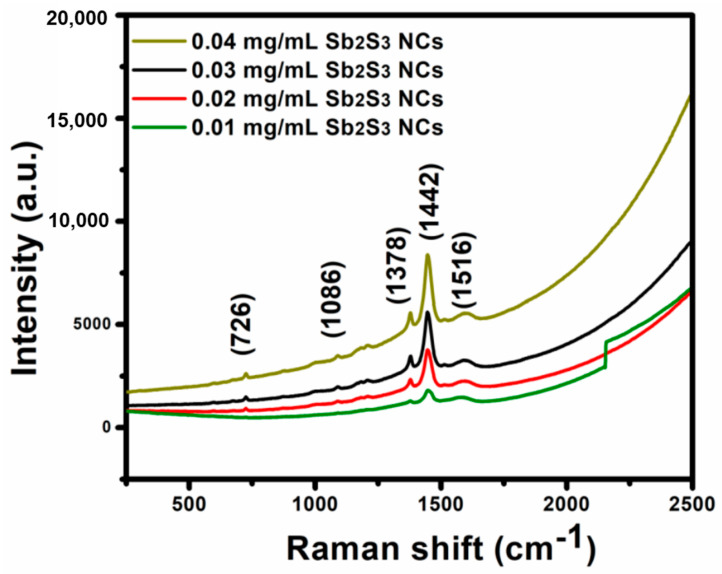
Raman spectra of theP_3_HT:PCBM:Sb_2_S_3_ NC blended thin films prepared using spin coating.

**Figure 4 polymers-13-02152-f004:**
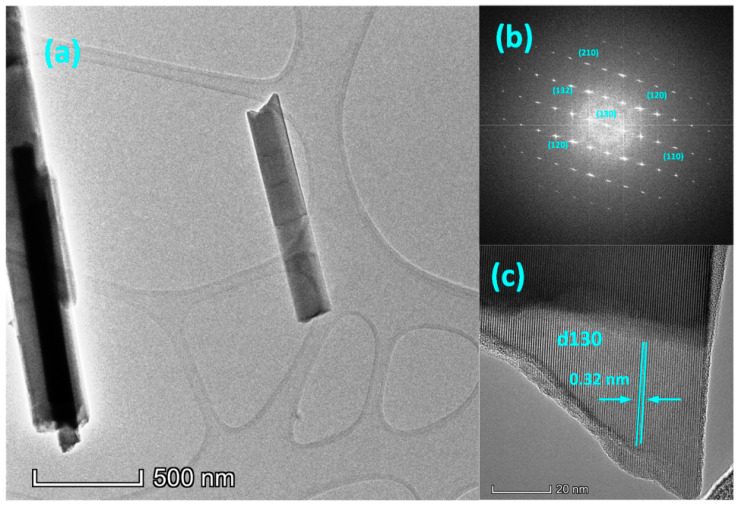
(**a**) TEM images of the Sb_2_S_3_ NCs. (**b**) Selective area electron diffraction (SAED). (**c**) High-resolution (HRTEM) image of the fabricated Sb_2_S_3_ NCs.

**Figure 5 polymers-13-02152-f005:**
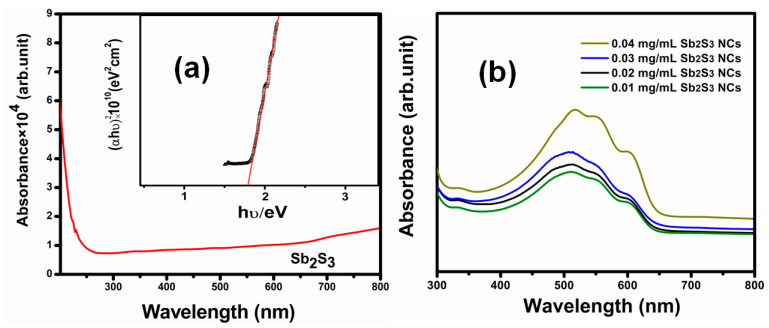
(**a**) UV–visible spectra absorption spectroscopy and band gap energy estimation of the Sb_2_S_3_ NCs (insert). (**b**) UV–visible spectra of theP_3_HT:PCBM:Sb_2_S_3_ NC polymer fractions blended with different Sb_2_S_3_ NC concentrations.

**Figure 6 polymers-13-02152-f006:**
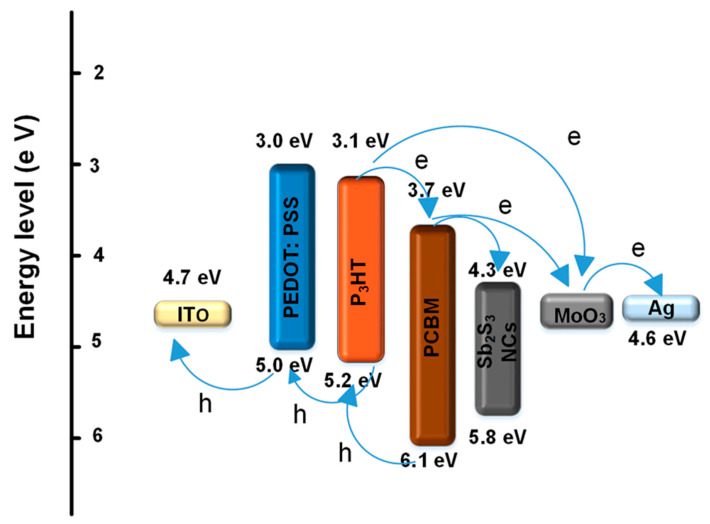
Energy band diagram and charge transfer effects of the P_3_HT, PCBM, and Sb_2_S_3_ NCs.

**Figure 7 polymers-13-02152-f007:**
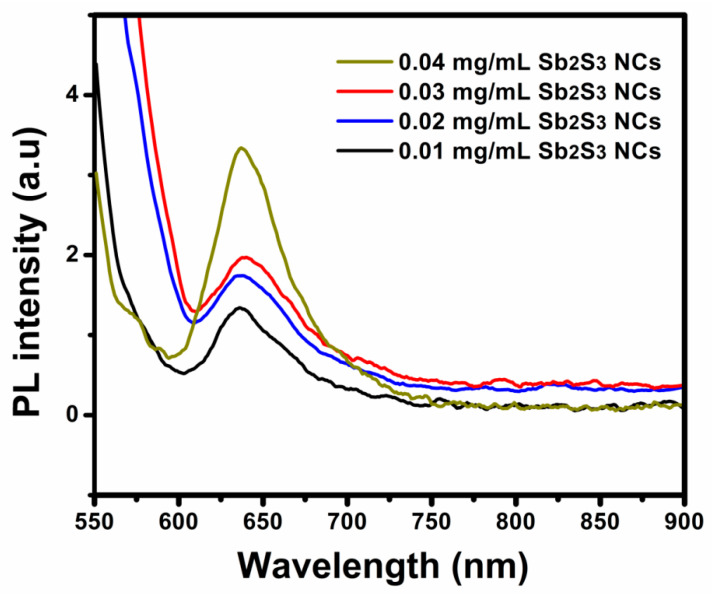
Photoluminescence spectra of the spin-coated P_3_HT:PCBM:Sb_2_S_3_ NCs prepared using different Sb_2_S_3_ NC concentrations.

**Figure 8 polymers-13-02152-f008:**
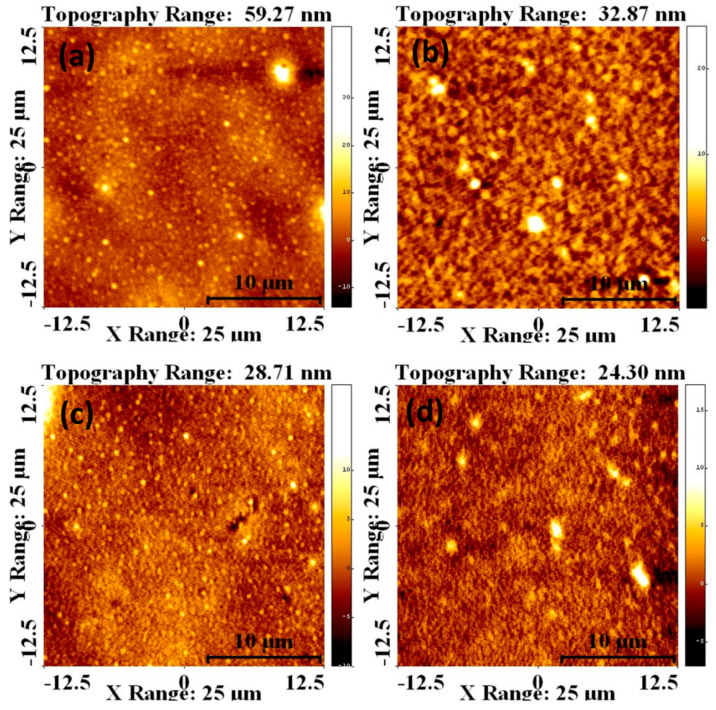
AFM images of the top view of the P_3_HT:PCBM:Sb_2_S_3_ NC blends prepared with Sb_2_S_3_ NC concentrations of (**a**) 0.01, (**b**) 0.02, (**c**) 0.03, and (**d**) 0.04 mg/mL.

**Figure 9 polymers-13-02152-f009:**
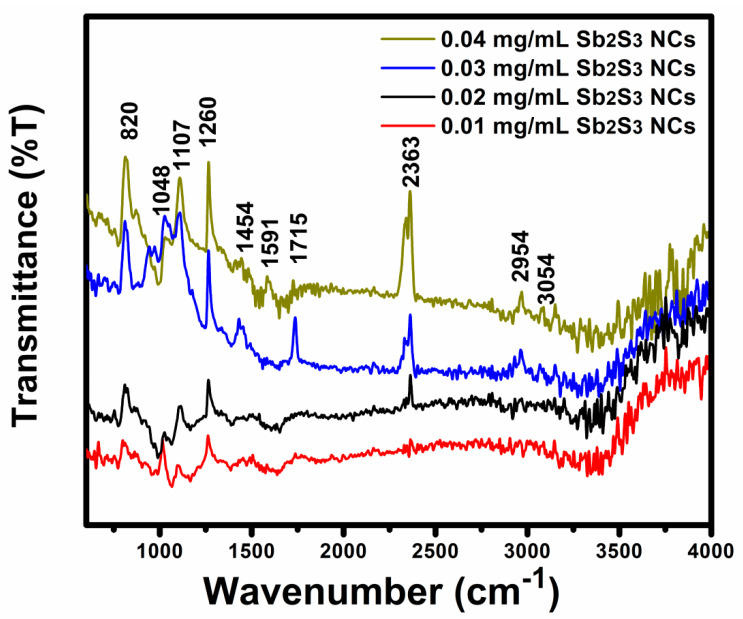
FTIR spectra of the P_3_HT:PCBM:Sb_2_S_3_ NC blended films with different Sb_2_S_3_ NC concentrations.

**Figure 10 polymers-13-02152-f010:**
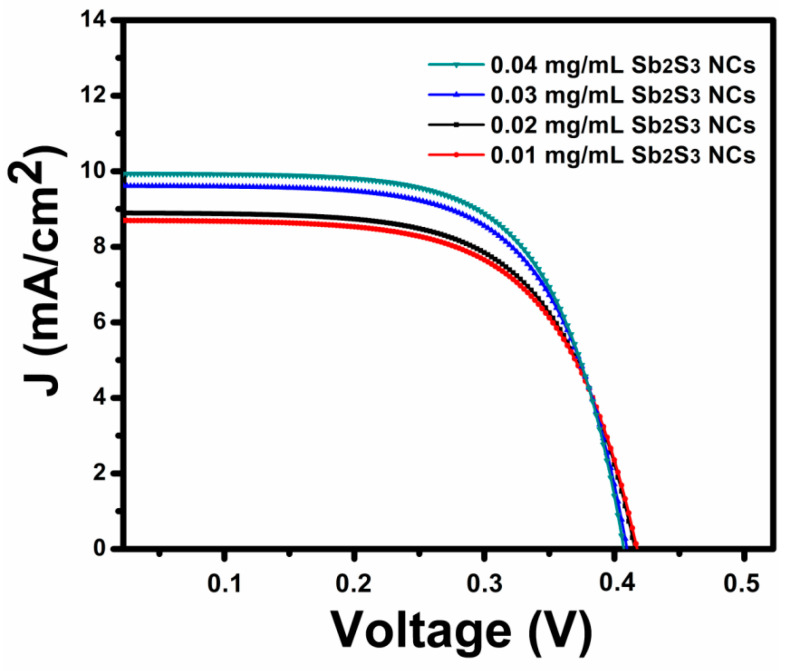
Current density–voltage (J–V) characteristics of the solar cell device with a glass/ITO/PEDOT:PSS/P_3_HT:PCBM:Sb_2_S_3_:NC/MoO_3_/Ag configuration.

**Table 1 polymers-13-02152-t001:** Detailed photovoltaic parameters of the organic solar cell device under AM 1.5 sun illumination with light power intensity of 100 mW/cm^2^.

Sample	V_oc_ (mV)	J_sc_ (mA/cm^2^)	FF (%)	η (%)	$R_{s}\;\left(Ω\;cm^{2}\right)$	$R_{\mathrm{sh}}\left(Ω\;cm^{2}\right)$
0.01 mg/mL	423	8.67	59.8	2.18	47.7	489.7
0.02 mg/mL	421	8.91	62.6	2.34	33.6	552.6
0.03 mg/mL	416	9.57	64.4	2.56	26.6	742.1
0.04 mg/mL	412	10.04	66.0	2.72	23.2	945.2

## Data Availability

The data presented in this study are available on request from the corresponding author.

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
