# Peer review of "Optimization of Sb2S3 Nanocrystal Concentrations in P3HT: PCBM Layers to Improve the Performance of Polymer Solar Cells"

_polymers, 2021, doi:10.3390/polym13132152_

Round 1
Reviewer 1 Report
In this manuscript, polymer solar cells were synthesized by adding Sb2S3 nanocrystals (NCs) to thin blended films with polymer poly(3-hexylthiophene) (P3HT) and [6,6]-phenyl-C61-butyric- acid-methyl-ester (PCBM) as p-type material was prepared via the spin-coating method. The effect of the Sb2S3 nanocrystals concentrations (0.01, 0.02, 0.03, and 0.04 mg/mL) in the polymer active layer was determined using different characterization instruments. Introducing Sb2S3 NCs increased light harvesting and regulated energy levels, improving the electronically parameters. Further, the absorber layer’s doping concentration played a definitive role in improving the device performance. This paper provided some valuable information and the content is very significant in this field. However, I recommended a major revision of the article from its present form before it can be published in polymers. Some specific comments are as follows:
- There are many reports on polymer solar cell applications. What are the advantages of this study? Please list the key comparison results.
- There are some grammar errors and inappropriate expressions.
- In the introduction part, the authors should explain the significance of the present report.
- The schematic representation is unclear. It needs more clarification.
- The authors should provide good images of SEM, TEM, and EDS data.
- The application part needs more explanation.
Author Response
Response to Reviewers Comments
polymers-1211603
Manuscript Title: Optimization of Sb2S3 Nanocrystals Concentrations in P3HT:PCBM Layers to Improve the Performance of Polymer Solar Cells
Submitted to polymers
- There are many reports on polymer solar cell applications. What are the advantages of this study? Please list the key comparison results.
By adding Sb2S3 NCs to polymer P3HT-PCBM thin blended films:
-Our results demonstrated that the electrons and holes transfer in the polymer active layer has enhanced, which led to improve the carrier separation efficiency and reduced the recombination.
-The growth of the P3HTis more aggregate and better the chains’ crystallinity and hole mobility.
- Absorption spectra of the sample after adding Sb2S3 NCs was better, which is attributed to the π-π* transitions.
- There is more improved phase separation size, which is important to the charge transport and collection.
Our results indicated that adding Sb2S3 NCs to polymer P3HT-PCBM has significant effectiveness in the polymer active layer properties, which can be an efficient method to improve thin film properties.
- There are some grammar errors and inappropriate expressions.
Answer 8: Authors thank the learned reviewer for the useful comment. We agree with the learned reviewer, English has been revised and corrected throughout the whole manuscript.
- In the introduction part, the authors should explain the significance of the present report.
Answer 2: Authors thank the learned reviewer for the useful comment. We agree with the learned reviewer. Accordingly, we added new information in the introduction section of the revised manuscript. Please see lines 86 and 91.
- The schematic representation is unclear. It needs more clarification.
We added this sentence in the revised manuscript (We agree with the learned reviewer. The processes can take place relying on the energy of excitation. When excitons are created upon light absorption in the Sb2S3 NCs, it is anticipated that the electrons shall be transferred to PCBM and holes to P3HT. While excitations are generated in P3HT, we can speculate from the energy levels that a transfer of electron shall take place towards Sb2S3 NCs and/or PCBM.) (Line 86 page 91).
- The authors should provide good images of SEM, TEM, and EDS data.
Answer 5: We agree with the learned reviewer. SEM, TEM, and EDS data are very important particularly as surface characterization methods. Therefore, we used Atomic force microscopy (AFM) to study the surface morphology of thin film. However, unfortunately, our scanning electron microscopy (SEM) equipment is not working. We apologize for that. Therefore, it is not possible to make further thin film characterizations at this stage.
- The application part needs more explanation.
Answer 6: Authors thank the learned reviewer for the useful comment.
We agree with the learned reviewer. Accordingly, we added new information in the introductionsection of the revised manuscript. (Line 1 page 42).
((Polymers are often used as adjustment additives in dye-sensitized solar cells to provide a desirably resilient substrate, a frame structure of the semi solid state electrolytes, in addition to the pore/film formation in the photoanode films. Moreover, polymers are added to enhance the solar cell device performance such as reinforcing the processes of crystallization and nucleation in the perovskite solar cell films. Polymers are applied as buffer layers or donor layers to improve the device efficiency. Polymers are also applied as electron transmitters, hole transmission materials, as well as interfacial layers which improve the carrier separation efficacy and minimize the recombination)).
Reviewer 2 Report
1)could the authors demonstrate briefly the disadvantages for the integrating of inorganic and nanostructures?
2) The author claimed that Sb2S3 can increase the light scattering to promote the increase of light absorption in the active layer. In the manu., the author have used the amount range of Sb2S3 from 0.01 mg/ml to 0.04 mg/ml. How is the performance when the amount increase more than 0.04 mg/ml?
3) In the optical absorption spectra, the abs. of theP3HT:PCBM:Sb2S3 NC increased from 400nm to 650 nm, however, there is almost no change from 700nm to 800 nm, Could the authors demonstrate the reasons? Besides, could the author provide the abs. of pure polymer film without Sb2S3 NCs?
4) The figures and icons are inconsistent, not alignment. The authors should update the figures. Besides, the format of article should be carefully checked.
Author Response
Response to Reviewers Comments
polymers-1211603
Manuscript Title: Optimization of Sb2S3 Nanocrystals Concentrations in P3HT:PCBM Layers to Improve the Performance of Polymer Solar Cells
Submitted to polymers
First of all, I would like in my capacity as the corresponding author to express my sincere gratitude and appreciation to the reviewers for their valuable comments without which this manuscript would not have been valuable. I am really greatly indebted to the reviewer and to the Editor. In the revised manuscript, we have accommodated all concerns of the reviewers.
Reviewer #1:
1)could the authors demonstrate briefly the disadvantages for the integrating of inorganic and nanostructures?
In recent years there have been significant efforts in the development of organic photovoltaic (OPV) cells, because they offer inexpensive alternative to silicon solar cells. In addition these cells are environmental friendly, light, semi-transparent, flexible and easy to produce and integrate. However, organic solar cells have some drawbacks, such as relatively low efficiency, lower conductivity and faster degradation in ambient conditions. It has been suggested that the addition of an inorganic acceptor material to form an organic–inorganic hybrid solar cell should theoretically improve the performance of OPV by enhancing the absorption and improving the charge transport characteristics. However, to date, the efficiency of hybrid solar cells have been very low in comparison with organic solar cells. Another disadvantage of the hybrid solar cells is their relatively short stability. There are several reasons determining the challenges for hybrid solar cells: (i) design considerations when choosing an inorganic material with appropriate band gap and a HOMO level offset to allow both a significant absorption contribution as well as a large Voc. (ii) nanoparticles aggregation and control the amount of nanoparticles for efficient percolating network formation (iii) nanoparticles are stabilized in solvents by ligands, which are insulating and negatively affect the electrical performance of film. A deeper understanding of these drawbacks, as well as the fundamental mechanisms determining the operation of hybrid solar cells is required to efficiently and effectively increase device performance. Our study aims to contribute to understanding the mechanism of operation of inorganic nanoparticles-organic hybrid solar cells.
2) The author claimed that Sb2S3 can increase the light scattering to promote the increase of light absorption in the active layer. In the manu., the author have used the amount range of Sb2S3 from 0.01 mg/ml to 0.04 mg/ml. How is the performance when the amount increase more than 0.04 mg/ml?
Answer 3: The reviewer is correct. Our experimental results indicated that an increase in the Sb2S3 NCs concentration to more than 0.04 mg/L was associated with the solution viscosity, which increased with the increase in the Sb2S3 NCs concentration, leading to the formation of many black agglomerates on the bottom of the beaker.
3) In the optical absorption spectra, the abs. of theP3HT:PCBM:Sb2S3 NC increased from 400nm to 650 nm, however, there is almost no change from 700nm to 800 nm, Could the authors demonstrate the reasons? Besides, could the author provide the abs. of pure polymer film without Sb2S3 NCs?
Answer 1: Authors thank the learned reviewer for the useful comment. Depending on previous studies, Sb2S3 NCs had absorption spectra below 550 nm and the absorption spectra of the main P3HT polymer in a range of 450-600 nm. This made the absorption spectra of the P3HT: PCBM: Sb2S3 NCs active layer showed an absorption just in range from 400nm to 650 nm, and no change from 700nm to 800 nm. Unfortunately, we did not fabricate pure P3HT:PCBM polymer thin film without Sb2S3 doping . We did more than 37 polymer thin films in this work to obtain these results.
4) The figures and icons are inconsistent, not alignment. The authors should update the figures. Besides, the format of article should be carefully checked.
Answer 9: Authors thank the learned reviewer for the useful comment. We agree with the learned reviewer. Accordingly, we replaced fig.6 and fig.7. We corrected figures and icons and the format throughout the manuscript in the revised manuscript. The present format is the journal’s format.
We appreciate the reviewer’s feedback and valuable suggestions without which, this manuscript would not have been so well organized and significantly improved.
Reviewer 3 Report
M. Mkawi et al. showed the manuscript titled ‘Optimization of Sb2S3 Nanocrystals Concentrations in P3HT: PCBM Layers to Improve the Performance of Polymer Solar Cells’, and it contains nanocrystals and Device Fabrication, Characterization such as XRD diffraction, TEM, AFM, FTIR, Raman spectroscopy UV-visible spectra absorption spectroscopy, Photoluminescence, J-V characteristics and an so on. It tries to explain the superior performance of Sb2S3NCs doped P3HT: PCBM. This title has a certain interests in the field of chemistry, energy, and materials. But there are still some problems that need to be solved by the authors to improve the quality of the manuscript.
The specific questions are as follows:
- In line 183, please check the sentence ‘Raman spectroscopy in a range of 250 to 500 cm-1 was used to investigate the mole-183 cules’ irrational mode in the P3HT: PCBM:Sb2S3 NCs blended films.’ Especially for expression of Raman spectroscopy in a range. Why is 250 to 500 cm-1?
- In the introduction, the authors state,‘The charge carrier transfer across the P3HT-PCBM:Sb2S3 NCs was also assessed using Raman spectroscopy measurements in the active layer’. But this discussion of lin 183-202, author could not explain Raman spectroscopy evidence to assess charge carrier transfer. Why? This part lacks the corresponding analysis and explanation.
- In UV-visible spectra of theP3HT:PCBM:Sb2S3 NC polymer, the author should explain why absorption strength was increased along with the different concentrations increase of Sb2S3 NCs (from 0.01 to 0.04 mg/mL)? What are the physical and chemical meanings of the increased absorption strength?
- From Figure 7. Energy band diagram, the author states there is three charge transfer channel in the P3HT:PCBM:Sb2S3 NC system, such as (a) electron transport from the P3HT to the PCBM, (b) electron transfer from the P3HT to the Sb2S3 NCs, and (c) electrons moving from the P3HT to the Sb2S3 NCs and then to the PCBM.
*About this, the first question is if the (c) electrons moving from the P3HT to the Sb2S3 NCs and then to the PCBM can take place because energy level of PCBM is higher than Sb2S3 NCs?
*Second questions: According to the principle of heterojunction cell, charge separation is expected to occur at the P3HT: PCBM interface. Generally, doping Sb2S3 NCs can enhance charge separation; however, more evidence should be provided to support the possible charge transfer channels (a-c, three possible reasons ). Please add the necessary reference and discussion.
* It is necessary for the author to explain what factors play a leading role in your research system. But at present, the author's analysis is not comprehensive.
Author Response
Response to Reviewers Comments
polymers-1211603
Manuscript Title: Optimization of Sb2S3 Nanocrystals Concentrations in P3HT:PCBM Layers to Improve the Performance of Polymer Solar Cells
Submitted to polymers
- In line 183, please check the sentence ‘Raman spectroscopy in a range of 250 to 500 cm-1 was used to investigate the mole-183 cules’ irrational mode in the P3HT: PCBM:Sb2S3 NCs blended films.’ Especially for expression of Raman spectroscopy in a range. Why is 250 to 500 cm-1?
Answer 1: The reviewer is correct. It should be (Raman spectroscopy in the range of 250 to 2500 cm-1).Accordingly, we corrected this information in the revised manuscript. It is an editorial mistake. (Please see Line 160 page 5).
2.In the introduction, the authors state, ‘The charge carrier transfer across the P3HT-PCBM:Sb2S3 NCs was also assessed using Raman spectroscopy measurements in the active layer’. But this discussion of line 183-202, author could not explain Raman spectroscopy evidence to assess charge carrier transfer. Why? This part lacks the corresponding analysis and explanation.
Answer 2: The reviewer is correct. We deleted this sentence in the revised manuscript. It is an editorial mistake.
3.In UV-visible spectra of theP3HT:PCBM:Sb2S3 NC polymer, the author should explain why absorption strength was increased along with the different concentrations increase of Sb2S3 NCs (from 0.01 to 0.04 mg/mL)? What are the physical and chemical meanings of the increased absorption strength?
Answer 3: The absorption strength was increased along with the different concentrations increase due to improvements in polymer crystallinity. The crystallinity affects the optical, thermal, mechanical, and chemical properties of the polymer.
We added this sentence in the revised manuscript (The absorption strength was increased along with the different concentrations increase due to improvements in polymer crystallinity) (Please see Line 211 page 16).
4.From Figure 7. Energy band diagram, the author states there is three charge transfer channel in the P3HT:PCBM:Sb2S3 NC system, such as (a) electron transport from the P3HT to the PCBM, (b) electron transfer from the P3HT to the Sb2S3 NCs, and (c) electrons moving from the P3HT to the Sb2S3 NCs and then to the PCBM.
*About this, the first question is if the (c) electrons moving from the P3HT to the Sb2S3 NCs and then to the PCBM can take place because energy level of PCBM is higher than Sb2S3 NCs?
Answer 4-1: The reviewer is correct. The processes can occur depending on the excitation energy. When excitons are formed upon light absorption in the Sb2S3 NCs, it is expected that the electrons will be transferred to PCBM and holes to P3HT. When excitations are generated in P3HT, we can predict from the energy levels that an electron transfer will occur towards Sb2S3 NCs and/or PCBM. To learn more about the processes occurring between the blend components, we investigated steady state and time-resolved PL
We added this sentence in the revised manuscript
(The processes can occur depending on the excitation energy. When excitons are formed upon light absorption in the Sb2S3 NCs, it is expected that the electrons will be transferred to PCBM and holes to P3HT. When excitations are generated in P3HT, we can predict from the energy levels that an electron transfer will occur towards Sb2S3 NCs and/or PCBM. To learn more about the processes occurring between the blend components, we investigated steady state and time-resolved PL). (Line 8 page 222).
*Second questions: According to the principle of heterojunction cell, charge separation is expected to occur at the P3HT: PCBM interface. Generally, doping Sb2S3 NCs can enhance charge separation; however, more evidence should be provided to support the possible charge transfer channels (a-c, three possible reasons ). Please add the necessary reference and discussion.
Answer 4-2: We agree with the learned reviewer’s comment. We added new references supporting the possible charge transfer channels in the revised manuscript. It was an editorial mistake. (Line 8 page 222,26).
* It is necessary for the author to explain what factors play a leading role in your research system. But at present, the author's analysis is not comprehensive.
Answer 4-3: We agree with the learned reviewer’s comment.
Efficient bulk heterojunction (BHJ) polymer solar cells (PSCs) based on P3HT:PCBM:Sb2S3: NCs were fabricated by choosing the processing parameters. The thickness of the active layer was found to be about 100-120 nm, and the spin-coated layer was annealed at 140oC for 10 min. The effect of cathode interfacial layers on device performance is related to the formation of interfacial dipoles. Furthermore, the effect of PEDOT: PSS layer interfacial (thickness ~60 nm annealing at 120oC for 15 min) on device performance is attributed to good interfacial conductivity and its optical property. The metal electrode and a buffer layer deposited in the slow rate have a better influence on device performance. Additionally, multiple factors in solar cell design play roles in limiting the cell's ability to convert the sunlight it receives. Such as sunlight wavelength, charge carrier recombination, solar cells temperature, light reflection
Round 2
Reviewer 1 Report
The manuscript can be acceptable in the present form.
Reviewer 2 Report
I am satisfied with the revisions that the authors have made. The manuscript can be accepted for publication as it is.
Reviewer 3 Report
The authors have made changes, and the manuscript can be accepted.
This manuscript is a resubmission of an earlier submission. The following is a list of the peer review reports and author responses from that submission.
Round 1
Reviewer 1 Report
Mkawi et al. reported the incorporation of Sb2S3 Nanocrystals in P3HT:PCBM layers to improve the performance of polymer solar cells. At optimized Sb2S3 concentration of 0.04 mg/mL, the P3HT:PCBM based solar cell reaches the power conversion efficiency of 2.72%. However, I think that this research is not timely. First of all, the active material of P3HT:PCBM is no longer promising material for OPV because there are several conjugated polymer doners that give higher efficiency than P3HT while fullerene-based acceptors have been less interested in research community due to their limited absorption. Moreover, the performance of OPV needs to improve substantially; the PCE of optimized device (2.72%) is much lower than control device (without Sb2S3) reported in the literatures. Especially, Voc of 0.4V is very low, which shows the serious recombination within the device. Therefore, I do not recommend to publishing this article at Polymer. However, authors may consider the sister journals form MDPI such as Photonics, Applied Science. Some comments to improve the manuscript are:
(1) The device data of P3HT:PCBM (without Sb2S3) should be added.
(2) Voc decreases with higher concentration of Sb2S3 although PCE increases. Author should find out the reason behind low Voc with higher concentration of Sb2S3. Since Voc tells the quality of a solar cell, improving the Voc is important.
(3) Because PCE keeps increasing with higher concentration of Sb2S3, the concentration of 0.04mg/L may not be optimized one. Higher concentration should be explored until reaching the peak PCE.
(4) From experimental point of view, 0.01 mg is very small. Even 1 mg is very difficult to precisely weighed. Authors should clarify how the extremely small amount (0.01 mg) of Sb2S3 is precisely weighed and controlled during the experiment.
(5) EQE data should be provided.
(6) PL spectrum shown in Fig. 6 seems not correct. It looks like absorption spectrum.
(7) In Fig 5 (b), with very little concentration of Sb2S3, the absorbance of the film should not be different so much.
(8) In introduction, “They are considered the best fullerene derivative-based donor-accepter electron material for organic solar cells” should be revised since P3HT:PCBM is not the best donor:acceptor system.
(9) Authors should carefully check the typo error throughout the manuscript. For example, “Trancmittance” in Y-axis of Fig 9. In the unit (Cm-1), “C” should be small letter.
Reviewer 2 Report
In this manuscript, polymer solar cells were synthesized by adding Sb2S3 nanocrystals (NCs) to thin blended films with polymer poly(3-hexylthiophene) (P3HT) and [6,6]-phenyl-C61-butyric- acid-methyl-ester (PCBM) as p-type material was prepared via the spin-coating method. Introducing Sb2S3 NCs increased light-harvesting and regulated energy levels, improving the electronic parameters. Further, the absorber layer’s doping concentration played a definitive role in improving the device performance. This paper provided some valuable information and the content is very significant in this field. However, I recommended a major revision of the article from its present form before it can be published in polymers. Some specific comments are as follows:
- The abstract and conclusion sections should be a specific and scientific approach.
- In the introduction section, the authors should expound the research significance of the present work.
- The authors should explain the novelty of the present report?
- What is the pH of the reaction solution? The pH of the solution normally varies from precursor to precursor. The authors must justify the selection of pH, temperature, and time in the experimental procedure.
- The morphology studies are unclear. The authors should provide good images of SEM, TEM, and EDS data.
- The authors should explain the application part in detail.
- The obtained should compare with the literature.
- In the current state, there are more typographical errors and the language should be improved. Therefore, the authors are advised to recheck the whole manuscript for improving the language and structure carefully.